# Health and economic benefits of secondary education in the context of poverty: Evidence from Burkina Faso

**Luisa K. Werner**[1,2]*, **Jan-Ole Ludwig**[1,3], **Ali Sie**[4], **Cheik H. Bagagnan**[4], **Pascal Zabré**[1,4], **Alain Vandormael**[1], **Guy Harling**[5,6,7,8,9], **Jan-Walter De Neve**[1‡], **Günther Fink**[10,11‡]

**1** Heidelberg Institute of Global Health, Medical Faculty and University Hospital, University of Heidelberg, Heidelberg, Baden-Württemberg, Germany, **2** Medical Faculty and University Hospital, University of Freiburg, Freiburg im Breisgau, Baden-Württemberg, Germany, **3** Medical Faculty and University Hospital, University of Heidelberg, Heidelberg, Baden-Württemberg, Germany, **4** Health Research Centre of Nouna, Ministry of Health, Nouna, Kossi Province, Boucle du Mouhoun Region, Burkina Faso, **5** Department of Epidemiology, Harvard T. H. Chan School of Public Health, Boston, Massachusetts, United States of America, **6** Harvard Center for Population and Development Studies, Harvard University, Cambridge, Massachusetts, United States of America, **7** Africa Health Research Institute, Congella, Durban, South Africa, **8** Institute for Global Health, University College London, Royal Free Hospital, London, United Kingdom, **9** MRC/Wits Rural Public Health and Health Transitions Research Unit, University of Witwatersrand, Agincourt, Bushbuckridge Local Municpality, Johannesburg, South Africa, **10** Swiss Tropical and Public Health Institute, Basel, Basel-Stadt, Switzerland, **11** University of Basel, Basel, Basel-Stadt, Switzerland

‡ JWDN and GF are joint senior authors on this work.
* luisa.katharina@gmx.de

**Data Availability Statement:** This was a complete case analysis, and all analyses were conducted in Stata MP 15.1 (StataCorp; College Station, TX). HDSS data are available in form of a limited dataset via https://doi.org/10.7910/DVN/1OITCD; for more

## Abstract

Even though formal education is considered a key determinant of individual well-being globally, enrollment in secondary schooling remains low in many low- and middle-income countries, suggesting that the perceived returns to such schooling may be low. We jointly estimate survival and monetary benefits of secondary schooling using detailed demographic and surveillance data from the Boucle du Mouhoun region, Burkina Faso, where national upper secondary schooling completion rates are among the lowest globally (<10%). We first explore surveillance data from the Nouna Health and Demographic Surveillance System from 1992 to 2016 to determine long-term differences in survival outcomes between secondary and higher and primary schooling using Cox proportional hazards models. To estimate average increases in asset holdings associated with secondary schooling, we use regionally representative data from the Burkina Faso Demographic Health Surveys (2003, 2010, 2014, 2017–18; $N$ = 3,924). Survival was tracked for 14,892 individuals. Each year of schooling was associated with a mortality reduction of up to 16% (95% CI 0.75–0.94), implying an additional 1.9 years of life expectancy for men and 5.1 years for women for secondary schooling compared to individuals completing only primary school. Relative to individuals with primary education, individuals with secondary or higher education held 26% more assets (SE 0.02; CI 0.22–0.30). Economic returns for women were 3% points higher than male returns with 10% (SE 0.03; CI 0.04–0.16) vs. 7% (SE 0.02; CI 0.02–0.012) and in rural areas 20% points higher than in urban areas with 30% (SE 0.06; CI 0.19–0.41) vs. 4% (SE 0.01; CI 0.02–0.07). Our results suggest that secondary education is associated with substantial health and economic benefits in the study area and should therefore be considered

extensive data contact the Nouna Health Research Center, Burkina Faso (CRSN, info@crsn-nouna.bf), and DHS survey data are available from the DHS Program (https://dhsprogram.com/). We do not report any special privileges for access of DHS data. Access of CRSN data was facilitated as there is a partnership among institutes and co-authors from the CRSN collaborating for this article. All data was fully anonymized before accessing.

**Funding:** This work was supported by the Alexander von Humboldt Foundation (https://www.humboldt-foundation.de/), funded by Germany's Federal Ministry of Education and Research (https://www.bmbf.de/en/index.html). JWDN was also supported by the European Commission (825823; https://ec.europa.eu/); German Science Foundation (405898232; https://www.dfg.de/en/); NICHD of NIH (R03-HD098982; https://www.nichd.nih.gov/); and the Heidelberg University Excellence Initiative (https://www.uni-heidelberg.de/excellenceinitiative/). GH was supported by a fellowship from the Wellcome Trust and Royal Society (210479/Z/18/Z). For the publication fee we acknowledge financial support by Deutsche Forschungsgemeinschaft within the funding programme „Open Access Publikationskosten" as well as by Heidelberg University. The funders had no role in study design, data collection and analysis, decision to publish, or preparation of the manuscript.

**Competing interests:** The authors have declared that no competing interests exist.

by researchers, governments, and other major stakeholders to create for example school promotion programs.

## Introduction

Education is widely considered as one of the main determinants of socioeconomic status yielding large benefits for the individual [1] as well as on a community level [2, 3]. Education is not only a key predictor of income [4] but has also been linked to many aspects of health. In low-resource settings, increased secondary schooling has been shown to lower HIV-infection risk [5], reduce child mortality [6, 7], and increase adult life expectancy [6–8]. In high-income settings, increases in compulsory education reforms have been shown to reduce adult mortality [9], and to increase life expectancy [10].

While substantial improvements in primary schooling have been achieved during the MDG era [11], the ambitious secondary school enrolment targets specified within the Sustainable Development Goals (SDGs) seem hard to achieve in some parts of the world [12]. Secondary schooling participation rates remain particularly low in sub-Saharan Africa, where only 42% enter last grade of lower secondary education [13]. School dropout may occur due to economic [14, 15] and social reasons [16]; as well as "external"—e.g. security issues, distance to school–and/or "internal" factors—e.g. poor school quality [17]. Even though the health and economic returns to secondary education in low-income settings are generally considered high [6–8], perceptions of low or negative returns to schooling, particular in poor rural areas, may be an additional reason why adolescents discontinue formal education [18–20].

In this paper, we estimate economic and survival benefits of secondary education vs. primary education in poor rural areas of sub-Saharan Africa using detailed demographic surveillance data from Burkina Faso. Rather than just focusing on economic returns to schooling, we assess both the links between schooling and life expectancy, and the links between schooling and asset holdings. We study this question in a particular context, where upper secondary schooling completion rates are among the lowest globally (< 10% in 2013/14) [21]. To do so, we explore almost 25 years of surveillance data from the Nouna Health and Demographic Surveillance System (HDSS), in rural Burkina Faso, to determine long-run differences in survival outcomes across educational attainment levels. In a second analysis, we estimate economic benefits from schooling using multiple rounds from the Burkina Faso Demographic Health Surveys (DHS) data, and subsequently compare the health and wealth benefits of increased education in the region.

### Study context

This study was conducted in Boucle du Mouhoun region, north-west of Burkina Faso. The school system in Burkina Faso consists of a "6-4-3-system" with 6 years of primary, 4 years of lower secondary and 3 years of upper secondary schooling. Burkina's upper secondary school completion rate (<10% in 2013/14) is among the lowest in the world, with particularly low secondary schooling rates in rural areas and among girls [13, 21–23]. Burkina Faso's gross domestic product per capita (GDPPC) was USD 822 in 2019 (constant 2010 USD) [24]. The GDP is mainly created in the service sector (43.6%), one third comes from agriculture and one fifth from industry [25, 26]. In the Nouna HDSS in the region Boucle du Mouhoun most people live in rural areas and work in agriculture and animal husbandry [27]. The semi-urban town of Nouna has better access to the education and health system than the surrounding villages [27]. Life expectancy at birth in Burkina Faso rose from around 50 years in 2000 to 61 years in 2018 [28].

## Data description

### Nouna health and demographic surveillance survey data

The Nouna HDSS covers a population of 105,000 habitants with a mainly rural population [27, 29]. The Nouna HDSS is managed by the Health Research Centre of Nouna (Centre de Recherche en Santé de Nouna—CRSN) which has collected data since 1992 (initial census) and control census' in 1994, 1998 and 2009 [29]. The surveillance site was extended in 2000 and in 2004, now counting 58 villages vs. 39 in the initial census and the city of Nouna [29, 30]. Vital events were collected up to 3 times (every 120 days) per year besides other information such as verbal autopsies and household questionnaires among others [29]. The sampling strategy for the collection of information on educational status could not be reconstructed; information on educational status was given for almost one third of the total sample. See **S1 Table** for missingness. Additional details on the Nouna HDSS, data collection, and cohort are available elsewhere [27, 29–31]. For the current analysis, we extracted data on sex, age, education, survival status, time of visit, and migration status. Individuals entered the study when first observed and exited in case of death, migration out of the Nouna HDSS or other reasons. All individuals observed for at least 1 year (at least 2 observations) with complete data on variables of interest (age, sex, educational attainment, self-reported survival status) who were a (former) resident of the Nouna HDSS in the period 1992 to 2016, and born in or before 1980 were included in our analysis. We limited our sample to those born before 1980 to study old-age survival. Our final sample included 14,892 individuals with a mean observation time of 11.7 years (range: 1.0–24.6 years).

### Demographic and Health Survey (DHS) data

To estimate the economic benefits of education, we used multiple Burkina Faso DHS surveys with data on sub-national administrative units (2003, 2010, 2014, and 2017–18). DHS are nationally representative household surveys and cover all regions of Burkina Faso. Additional details on the DHS surveys have been described elsewhere [32]. To improve comparability with the Nouna HDSS data, we only included individuals from the Boucle du Mouhoun region, where the Nouna HDSS is located [29]. Data were extracted on sex, age, education, household wealth assets, geographical region, sampling weights, date of interview, number of household members, type of place of residence (rural/urban), and marital status, using information from the DHS household recode, personal recode and individual recode files. The DHS did not collect information on income because a majority of the population in this low-resource setting engages in subsistence farming or depend on informal jobs with irregular incomes. We included all working-age individuals (defined as 25 to 64 years), with complete data on educational attainment in the household roster. Information on educational status was given for 3,924 out of 5,043 working age individuals. The last two Burkina Faso survey rounds were Malaria Indicator Surveys (MIS) which do not collect educational information for men. See **S1 Fig** for a study participant flow diagram. Sampling weights were used for the creation of asset quintiles as outlined in DHS (2021).

## Statistical analysis

Our analysis proceeded in two steps. First, we conducted a survival analysis using Cox proportional hazards regression models and the Nouna HDSS dataset. Second, we estimated wealth regression using a Mincer equation and DHS data. **S2 Fig** displays a conceptual framework underlying our analysis.

## Survival analysis

To determine the relationship between education and survival status, we employed a multivariable Cox proportional hazards model. The Cox proportional hazards model is a survival model that allows the hazard of a certain event (here: death) to change over time under the condition that the hazard ratios in between the different groups stay the same (proportional- hazards assumption) [33]. In our application, the Cox proportional hazards model describes the hazard of "all-cause mortality" (outcome) while assuming that the hazards in between the different schooling groups remain the same. In the HDSS, information on educational attainment was collected for a subsample of 14,892 individuals. Most individuals entered the HDSS between 1990 and 2000 –this entry in the HDSS constitutes the beginning of the survival analysis. Observation periods end (censoring occurs) when individuals migrated out of the HDSS or exited the study for other reasons. Our primary exposure of interest was educational attainment. Due to a limited number of individuals in the higher education categories we used education as continuous variable instead of categorizing educational attainment as in subsequent analyses. We also predicted life expectancy for three levels of educational attainment defined as either no schooling, at least some primary schooling (1 to 6 years of schooling), or at least some secondary schooling and higher (7 + years of schooling). Our analysis focused on the full sample ('basic model', with years of birth between 1900 and 1980) as well as three birth cohorts, including individuals born between (i) 1[st] January 1940 and 31[st] December 1949, (ii) 1[st] January 1950 and 31[st] December 1959, and (iii) 1[st] January 1960 and 31[st] December 1969. In all analyses, we controlled for sex. Additionally, we controlled for year of birth in our 'basic model'. We assessed survival benefits separately by urban (city of Nouna) vs. rural (surrounding villages).

## Wealth regression

We use asset ownership as a proxy for permanent income and estimate average increases in asset holdings associated with additional education. Asset quintiles were used as outcome in standard Mincer regression models to estimate returns to education [34–36]. The Mincer equation expresses log earnings as a function of years of schooling and a quadratic labor market experience term. We abstained from including more controls in the equation as this might exacerbate bias already existent in the data [34, 37]. We categorized educational attainment into three groups: no schooling (<1 year), primary schooling (1 to 6 years of schooling), and secondary schooling and higher (7 and more years of schooling). Individuals having at least started primary school were chosen as reference group to compare income benefits associated with secondary schooling to those associated with primary schooling. We provide results for a continuous education variable (years) in **S2 Table**. Wealth regressions were performed for all working age individuals (25 to 64 years of age) with complete information on education. 1,119 working age individuals had no data on education leaving an analytical sample of 3,924 individuals for the Boucle du Mouhoun region. We adjusted for age, as a proxy for labor market experience, age squared as part of the Mincer equation and DHS survey year. For methodology and estimation of predicted incomes and relative health and financial returns see **S1 File**.

## Results

### Sample description HDSS data

As summarized in **Table 1**, a total of 14,892 individuals were included in the survival analysis. 51.2% were men and 47.9% were women. Median age of women was around one year higher than that of men when first enrolled in the HDSS (33 compared to 32 years). Median years of schooling was highest for men (3 years) ranging from 0–22 years compared to a median of 0

**Table 1. Baseline characteristics of the HDSS complete case dataset.**

| Subsample | Female | | | Male | | | Both sexes | | |
|---|---|---|---|---|---|---|---|---|---|
| | N or median | (IQR) | (% or range) | N or median | (IQR) | (% or range) | N or median | (IQR) | (% or range) |
| *Characteristic* | | | | | | | | | |
| Number of subsequent deaths | 843 | - | (11.8%) | 991 | - | (12.8%) | 1,834 | - | (12.3%) |
| Age at first visit (years) | 33 | (25–46) | (11–99) | 32 | (24–42) | (11–99) | 32 | (25–44) | (11–99) |
| Schooling (years) | 0 | (0–4) | (0–20) | 3 | (0–6) | (0–22) | 0 | (0–6) | (0–22) |
| Highest schooling attainment | | | | | | | | | |
| None (0 years) | 4,889 | - | (68.6%) | 3,541 | - | (45.6%) | 8,430 | - | (56.6%) |
| Primary (1–6 years) | 1,451 | - | (20.3%) | 2,560 | - | (33.0%) | 4,011 | - | (26.9%) |
| Secondary (7+ years) | 791 | - | (11.1%) | 1,660 | - | (21.4%) | 2,451 | - | (16.4%) |
| Observations | 7,131 | - | (47.9%) | 7,761 | - | (52.1%) | 14,892 | - | (100.0%) |

*Notes*: All 14,892 individuals were followed over the years 1992–2016. Data are from the Nouna Health and Demographic and Surveillance (HDSS), Burkina Faso. IQR: Interquartile range.

years for women (range: 0–20 years). Most individuals had no formal education (45.6% of men and 68.6% of women), with less than one fifth of individuals attaining 7 or more years of schooling (at least some secondary education). Characteristics of the 1940s, 1950s and 1960s cohorts are summarized in **S3 Table**.

## Survival benefits: Survival analysis

**Table 2** shows the results of our survival analysis using Cox proportional hazards models. Controlling for sex and year of birth we found that each year of school was associated with a 3.6% reduction in the mortality hazard (aHR 0.96, 95% Confidence Interval [CI] 0.95–0.98). Female mortality was generally lower than male mortality ('Basic model': aHR 0.73; CI 0.66–0.80). When we stratified results by decade of birth, largest mortality reductions were observed for the 1950–59 cohort (aHR 0.94; CI 0.91–0.98). We also found larger survival benefits for the

**Table 2. Survival analysis: Adjusted hazard ratios (95% CI) from multivariate Cox regression models with death as outcome variable.**

| Birth Cohort | All | 1940–49 | | | 1950–59 | | | 1960–69 | | |
|---|---|---|---|---|---|---|---|---|---|---|
| | Basic model | Total | Urban | Rural | Total | Urban | Rural | Total | Urban | Rural |
| **Variable** | | | | | | | | | | |
| Educational Attainment | 0.96*** | 0.96** | 0.95** | 0.87* | 0.94*** | 0.92*** | 0.96 | 0.96* | 0.96* | 0.84*** |
| | (0.95–0.98) | (0.92–1.00) | (0.91–0.99) | (0.75–1.01) | (0.91–0.98) | (0.88–0.97) | (0.89–1.03) | (0.93–1.00) | (0.92–1.00) | (0.75–0.94) |
| Female sex | 0.73*** | 0.59*** | 0.43*** | 0.8 | 0.60*** | 0.68** | 0.48*** | 0.64*** | 0.56*** | 0.7 |
| | (0.66–0.80) | (0.48–0.73) | (0.33–0.57) | (0.58–1.11) | (0.47–0.78) | (0.49–0.95) | (0.32–0.73) | (0.48–0.86) | (0.38–0.82) | (0.45–1.10) |
| Observations | 14,892 | 1,372 | 819 | 553 | 1,980 | 1,034 | 946 | 3,282 | 1,788 | 1,494 |

*Notes*: The 'basic model' includes years of birth 1900–1980. The sample of the three cohorts included 6,634 individuals. All individuals were followed over the years 1992–2016. Adjusted Hazard Ratios (aHR) from multivariable Cox regressions for the relationship between educational attainment (per additional year of schooling) and death (outcome variable). Confidence intervals in parentheses.

*** p<0.01

** p<0.05

* p<0.1.

In the basic model we controlled for year of birth. Data are from the Nouna Health and Demographic and Surveillance (HDSS), Burkina Faso. Sample sizes were low for rural areas in two cohorts.

**Table 3. Conditional life expectancy for individuals aged 23 to 114 years in years by sex and educational group.**

|  |  |  | **Male** | **Female** |
|---|---|---|---|---|
|  |  |  |  |  |
| Highest schooling attainment |  |  |  |  |
| none (0 years) |  |  | 70.8 | 74.7 |
| primary (1–6 years) |  |  | 72.6 | 79.5 |
| secondary and higher (7+ years) |  |  | 74.5 | 84.6 |
| Mean |  |  | 71.6 | 76.8 |
| Median |  |  | 70.8 | 74.7 |

*Notes*: Median life expectancy predicted from a pooled sample of individuals born between 1902 and 1969 from the Nouna Health and Demographic and Surveillance (HDSS) data, separately for men and women. Life expectancies reflect conditional probabilities among adults aged 23 to 114 in the Nouna HDSS data.

rural compared to the urban population with mortality reductions of 13.3% and 16.2% in the 1940–49 and 1960–69 cohorts ('Rural 1940–49': aHR 0.87, CI 0.75–1.01 and 'Rural 1960–69': aHR 0.84, CI 0.75–0.94; 'Urban 1940–49': aHR 0.95, CI 0.91–0.99 and 'Urban 1960–69': aHR 0.96, CI 0.92–1.00). **S3 Fig** shows exemplarily with Kaplan Meier curves that individuals with at least some schooling born in year 1940 and 1950 get older than individuals having received no schooling. **Table 3** shows median life expectancies for individuals aged 23 to 114 years by educational attainment level. Secondary- and higher-schooled men and women lived on average 1.9 and 5.1 years longer than primary-schooled men and women, respectively.

## Sample description DHS data

Sample characteristics for individuals from the Boucle du Mouhoun region surveyed in the DHS are shown in **Table 4**. 3,924 individuals were analyzed. 61.8% of respondents were female, with a slightly higher median age for men than for women (38 and 36 years respectively). Median schooling was 0 years with almost 80% of women and two thirds of men not having attended school. About one fifth of men and less than 10% of women went to secondary school. Median estimated yearly income was 883 USD with about USD 300 higher incomes for men than for women (USD 1,066 vs. USD 757).

## Economic benefits: Wealth regressions

**Fig 1** shows the basic relationship between schooling and increase in asset holdings. **Table 5** shows the results for the wealth regressions. On average, secondary or higher schooling was associated with an increase in asset holdings of 26% (SE 0.02; CI 0.22–0.30) in the pooled sample. Returns for women were 3% points higher than male returns: 10% (SE 0.03; CI 0.04–0.16) vs. 7% (SE 0.02; CI 0.02–0.012). The difference between rural and urban residents was even higher with more than 20% points difference: 30% (SE 0.06; CI 0.19–0.41) vs. 4% (SE 0.01; CI 0.02–0.07). Results were similar when analyzing the nationally representative dataset (**S4 Table**). For wealth regressions by year of educational attainment instead of educational attainment categories, see **S2 Table**. For results for Burkina Faso see **S4** and **S5 Tables**.

## Discussion

In this paper, we have used surveillance data from the Nouna HDSS and repeated cross-sectional data from the DHS to estimate the associations between secondary schooling, health, and incomes. We show that on average life expectancy for women with secondary or higher

**Table 4. Selected characteristics of the DHS sample, Boucle du Mouhoun region.**

| Subsample | Male | | | Female | | | Both Sexes | | |
|---|---|---|---|---|---|---|---|---|---|
| | N or median | (IQR) | (% or range) | N or median | (IQR) | (% or range) | N or median | (IQR) | (% or range) |
| *Characteristic* | | | | | | | | | |
| Age (years) | 38 | (30–47) | (25–64) | 36 | (30–45) | (25–64) | 37 | (30–45) | (25–64) |
| Schooling (years) | 0 | (0–6) | (0–18) | 0 | (0–0) | (0–18) | 0 | (0–3) | (0–18) |
| Highest schooling attainment | | | | | | | | | |
| none (0 years) | 900 | - | (60.0%) | 1,901 | - | (78.4%) | 2,801 | - | (71.4%) |
| primary (1–6 years) | 292 | - | (19.5%) | 295 | - | (12.2%) | 587 | - | (15.0%) |
| secondary (7+ years) | 307 | - | (20.5%) | 229 | - | (9.4%) | 536 | - | (13.7%) |
| Income estimate (USD) | 1,066 | (475–1,506) | (166–2,860) | 757 | (475–1,461) | (166–2,939) | 883 | (475–1,506) | (166–2,939) |
| Observations | 1,499 | - | (38.2%) | 2,425 | - | (61.8%) | 3,924 | - | (100.0%) |

*Notes*: Individuals were surveyed in the Burkina Faso Demographic and Health Surveys (DHS) of 2003, 2010, 2014, and 2017–18. Income was estimated based on each household's relative position in the wealth distribution of the country. No sampling weights were used for descriptive statistics. IQR: interquartile range. USD: Constant 2011 international US Dollar.

education is 5 years higher than life expectancy for women with only primary school; for men, the difference is about 2 years. We also show that the financial returns to secondary school are high, with an increase in asset quintiles of 26%. Benefits seemed to be generally higher for

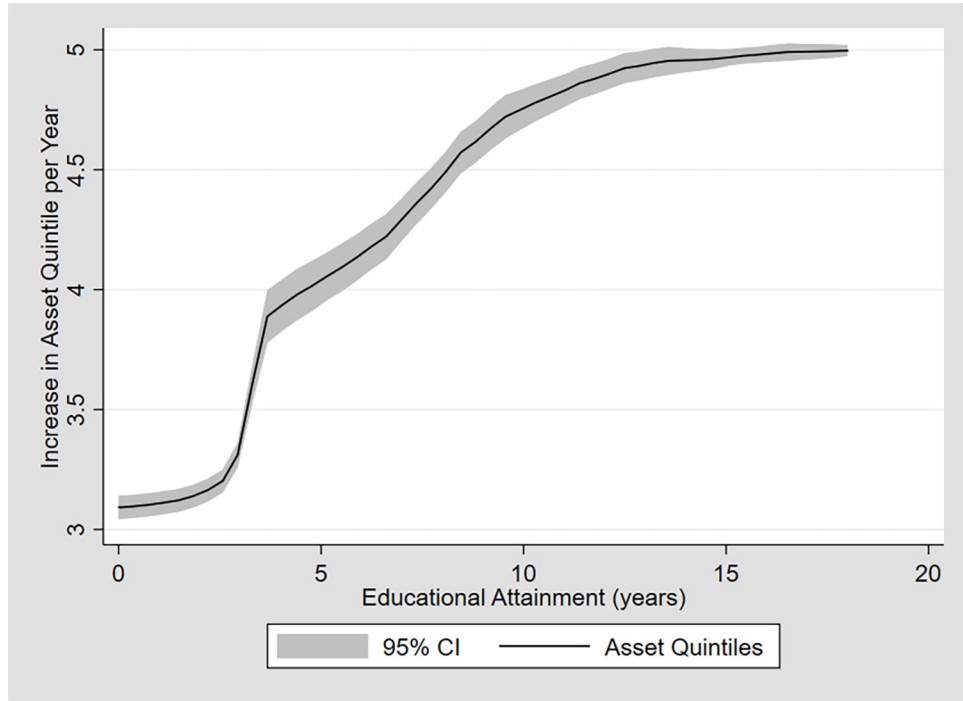

**Fig 1. Economic analysis.** Estimated increase in asset ownership by educational attainment level in the Boucle du Mouhoun region, Burkina Faso. *Notes*: Asset ownership by year of educational attainment based on OLS regression results from Mincer Earnings regressions, controlling for age, age squared and DHS survey year. Source: data for the Boucle du Mouhoun region using the Burkina Faso Demographic and Health Surveys (DHS) of 2003, 2010, 2014, and 2017–18 (N = 3,924). S4 Fig in the appendix shows results for the full (nationally representative) dataset of Burkina Faso.

**Table 5. Economic analysis: Results from OLS wealth regression models for the Boucle du Mouhoun region, Burkina Faso.**

| | | Basic Model | Stratified by Sex | | Stratified by Type of Place of Residence | |
|---|---|---|---|---|---|---|
| **Variable** | | | Males | Females | Urban | Rural |
| Age | | 0.01* | 0.02*** | 0.01* | 0.00 | 0.02*** |
| | | (0.01; 0.00–0.02) | (0.01; 0.01–0.04) | (0.01; -0.00–0.03) | (0.01; -0.01–0.01) | (0.01; 0.01–0.04) |
| Educational Attainment | | | | | | |
| No schooling | | -0.28*** | -0.15*** | -0.17*** | -0.06*** | -0.12*** |
| | | (0.02; -0.32 - -0.24) | (0.03; -0.21 - -0.10) | (0.03; -0.22 - -0.12) | (0.02; -0.09 - -0.02) | (0.03; -0.18 - -0.06) |
| Primary schooling | | Reference group | | | | |
| Secondary or higher | | 0.26*** | 0.07*** | 0.10*** | 0.04*** | 0.30*** |
| | | (0.02; 0.22–0.30) | (0.02; 0.02–0.12) | (0.03; 0.04–0.16) | (0.01; 0.02–0.07) | (0.06; 0.19–0.41) |
| Observations | | 3,924 | 1,499 | 2,425 | 1,449 | 2,475 |
| R-squared | | 0.15 | 0.38 | 0.27 | 0.25 | 0.04 |

*Notes*: A total of 3,924 individuals were surveyed in 2003, 2010, 2014, and 2017–18. The dependent variable was Ln (asset quintile). Coefficients represent the yearly increase in asset ownership on a natural logarithmic scale. Robust standard errors (SE) and 95% Confidence Interval (CI) in parentheses (SE; CI).

*** $p < 0.01$

** $p < 0.05$

* $p < 0.1$. In all models we controlled for age squared; in the stratified models we additionally controlled for survey round. OLS: ordinary least squares.

women and rural areas than those for men and urban areas. The higher returns to secondary education could be interpreted as evidence for education being particularly productive in groups where such education is scarce. It is also possible, that the additional educational opportunities after secondary schooling–such as vocational training–may be particularly attractive for women [38]. The relatively high returns to secondary education may of course also represent marriage outcomes, with more highly educated women much more likely to marry somebody with high socioeconomic status.

Our results suggest that both–wealth and health benefits—clearly make a major contribution to overall improvements in livelihoods. Only focusing on wealth gains will clearly underestimate the total benefits of staying in school.

The results presented here build on recent research showing that average rates of return are high in general and especially in sub-Saharan Africa (e.g., 12,5% in Montenegro et al, 2014), and that adolescents enrolled in school have a wide range of improved health outcomes compared to their peers in this setting [4, 39, 40]. Concerning the magnitude of health benefits, Gathmann, Jürges et al. (2015) estimate a 2.0% reduction in 20-year-mortality rate for only men and Lleras-Muney (2005) [41] a 10-year mortality rate reduction by 1.3 to 3.6% per additional year of education. Those results are similar to our estimations finding a 3.6% reduction in the mortality hazard. Perceived returns to education are high among other key stakeholders in the demographic surveillance area, such as parents, teachers, as well as health workers [42]. There appears to be a general consensus in literature that women receive higher economic benefits than men (for example an average rate of return of 14,6% for women vs, 11,3% for men) [4, 8, 43, 44] in line with our findings. Peet et al. (2015) found lower benefits for rural areas contrary to our findings. Concerning returns to schooling according to schooling level, Barouni and Broecke (2014) found higher returns for higher (secondary and tertiary) schooling than for primary schooling similar to the current paper, while Psacharopoulos (1985) found highest returns for primary education in the past. This might show a tendency towards higher returns for higher schooling levels. Concerning the methodology, most studies on returns to schooling use the discounting method or Mincer regression similar to us [39].

Literature on returns to education is large–estimates from this setting are however rare [45, 46]. Most of these studies, however, focus on either survival [3] or economic benefits [47, 48]. To our knowledge, only one study to date examines both monetary and non-pecuniary returns to schooling, but in a high-resource setting [49].

A key strength of the analysis is the combination of 25 years of longitudinal data on adult survival with detailed cross-sectional data from the region. This link allows us to comprehensively analyze the long-term outcomes related to education. Future research should take into consideration different dimensions of returns to education, and not concentrate on economic returns alone. Our results suggest that secondary schooling holds significant potential for increasing general life expectancy and earnings and should therefore be considered by researchers, governments, and other major stakeholders to create for example school promotion programs with different incentives. Formal schooling as part of human capital empowers individuals and thereby societies and helps to translate knowledge into progress. We suggest replicating similar analyses across multiple sites (e.g., HDSS sites) to compare and corroborate our results. Further research will be needed to understand the main barriers to secondary schooling access in this low-resource setting.

## Study limitations

Our analysis has several limitations. First, data on educational attainment was not collected for all individuals, which means that the results presented may not necessarily be representative of the entire Boucle du Mouhoun community. Second, our life expectancy estimates do not represent life expectancy at birth but rather conditional life expectancies as adults and are thus not directly comparable to national life expectancy estimates. Given that secondary education becomes only relevant in adolescence, we believe that the focus on adult survival outcomes is well justified in our analysis. Third, we only had data on years of schooling, while other aspects of schooling, and schooling quality in particular, are likely also important [50]. Similarly, limited data was available in the HDSS data on other health outcomes beyond survival. Education may have additional health benefits, such as improved mental health, which may not be fully captured in our survival data. Conversely, our definition of "return" does not account for the short-term cost faced by adolescents and families as no data was available on expenditures linked to schooling. Fourth, our research design is observational by construction and does not allow causal inference. Fifth, endogenous selection bias cannot be excluded: Wealthy individuals for example might select into education and bias health and income benefits positively, or very intelligent individuals who might bias stronger income than mortality results.

## Conclusions

Using almost 25 years of surveillance data from the Nouna HDSS and cross-sectional data from the DHS, we found that the returns to secondary schooling in rural Burkina Faso are high both in terms of lifetime income and in terms of life expectancy. Women and men with secondary or higher schooling experienced a 26% increase in wealth quintiles and lived 2 to 5 years longer than individuals completing only primary education. Despite these large benefits, less than 10% of adolescents currently finish secondary schooling in this setting [21]. Further research is urgently needed to identify the most effective strategies for reaching the ambitious SDG goals for secondary schooling.

## Supporting information

**S1 File. Predicted incomes and relative health and financial returns to education.**
(ZIP)

**S1 Fig. Study participant flow diagram DHS sample Boucle du Mouhoun region.**
(DOCX)

**S2 Fig. Framework of statistical analysis.**
(DOCX)

**S3 Fig. Survival according to schooling—Kaplan Meier graphs.**
(DOCX)

**S4 Fig. Wealth analysis: Estimated income increase by educational attainment for entire Burkina Faso.**
(DOCX)

**S1 Table. Missingness table Nouna HDSS.**
(DOCX)

**S2 Table. Wealth regressions for Boucle du Mouhoun, Burkina Faso with education as continuous variable.**
(DOCX)

**S3 Table. Sample characteristics of cohorts followed over time.**
(DOCX)

**S4 Table. Wealth regressions for entire Burkina Faso with education as categorized variable.**
(DOCX)

**S5 Table. Earnings regressions for entire Burkina Faso with education as continuous variable.**
(DOCX)

## Acknowledgments

We appreciate assistance provided by the CRSN and the Heidelberg Institute of Global Health. We want to especially acknowledge the process of data collection done by the CRSN and the kind sharing of this data for our analysis.

## Author Contributions

**Conceptualization:** Jan-Walter De Neve, Günther Fink.

**Data curation:** Luisa K. Werner, Jan-Ole Ludwig, Cheik H. Bagagnan, Pascal Zabré.

**Formal analysis:** Luisa K. Werner, Jan-Ole Ludwig.

**Funding acquisition:** Jan-Walter De Neve.

**Investigation:** Ali Sie, Cheik H. Bagagnan, Pascal Zabré.

**Methodology:** Luisa K. Werner, Jan-Ole Ludwig, Günther Fink.

**Project administration:** Jan-Walter De Neve, Günther Fink.

**Resources:** Ali Sie, Cheik H. Bagagnan, Pascal Zabré.

**Supervision:** Jan-Walter De Neve, Günther Fink.

**Validation:** Jan-Walter De Neve, Günther Fink.

**Visualization:** Luisa K. Werner.

**Writing – original draft:** Luisa K. Werner.

**Writing – review & editing:** Jan-Ole Ludwig, Alain Vandormael, Guy Harling, Jan-Walter De Neve, Günther Fink.

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
