## [Decision Letter · Decision Letter 0]

15 Dec 2021

PONE-D-21-01735

Health and economic benefits of secondary education in the context of poverty: Evidence from Burkina Faso

PLOS ONE

Dear Dr. Werner,

Thank you for submitting your manuscript to PLOS ONE. After careful consideration, we feel that it has merit but does not fully meet PLOS ONE’s publication criteria as it currently stands. Therefore, we invite you to submit a revised version of the manuscript that addresses the points raised during the review process.

The two reviewers raised overlapping concerns about the study rationale and its background. Please address all reviewer comments, in particular requests for clarification regarding the data analysis.

We look forward to receiving your revised manuscript.

Kind regards,

Yann Benetreau, PhD

Senior Editor

*PLOS ONE*

Journal Requirements:

2. In ethics statement in the manuscript and in the online submission form, please provide additional information about the patient records used in your retrospective study. Specifically, please ensure that you have discussed whether all data were fully anonymized before you accessed them and/or whether the IRB or ethics committee waived the requirement for informed consent. If patients provided informed written consent to have data from their medical records used in research, please include this information.

Reviewers' comments:

Reviewer's Responses to Questions

**Comments to the Author**

1. Is the manuscript technically sound, and do the data support the conclusions?

Reviewer #1: Partly

Reviewer #2: No

2. Has the statistical analysis been performed appropriately and rigorously? 

Reviewer #1: No

Reviewer #2: Yes

3. Have the authors made all data underlying the findings in their manuscript fully available?

Reviewer #1: Yes

Reviewer #2: Yes

4. Is the manuscript presented in an intelligible fashion and written in standard English?

Reviewer #1: Yes

Reviewer #2: Yes

5. Review Comments to the Author

Reviewer #1: I think this is a very interesting paper on a very important topic. It is evident that there's a lot pf hard work behind this manuscript however, in some places, it seems that the authors assumed that the crucial aspects of their analysis are common knowledge. For instance, Cox methodology is widely used but this paper should at least provide a brief description about it. Similarly, datasets and their sampling methodologies are not explained in detail. I think it is very important to include this information.

From estimations it appears as if the variables reported in the tables are the only ones included in the regressions i.e there are no controls. This is potentially problematic and all confounding variables should be included. There are are several demographic variables in DHS that could be used.

Mincers equation also seems to be incomplete. Age squared in missing and authors should also attempt education squared in continuous form. See schooling locus in George Borjas book of labor economics.

Citations seem a little unusual to me. Authors mention study number for direct citations and mention the sane number in parentheses as well. Please check the convention.

It would've been awesome if authors explained the results in context of Burkina Faso. It would've been easier to grasp the meaning of 1000 USD in context of the country being studied. Storytelling is the weakest aspect of this paper. Too much emphasis is on estimations and even the review section seems to be rushed. However, with little polishing, it can be accepted.

Reviewer #2: This paper provides evidence about the health and economic benefits of secondary education in rural Burkina Faso, using two sources of evidence: long-run health surveillance data, and the DHS. They find evidence that attending secondary school is associated with higher income as well as higher life expectancy.

My overriding comment about this paper is that it provides only correlational evidence about the relationship of interest, when we know that correlational evidence is probably severely limited, and causal evidence is widely available. The authors' acknowledgment of this fact (absence of causal evidence) is limited to a single sentence in the limitations section. Particularly for the analysis of the returns to education, there is an absolutely enormous economics literature analyzing causal effects of education on earnings in the developing world already published, and so the bar for an additional contribution is high. The authors fail to cite the majority of this literature, I won't aim to summarize it myself but will refer the authors to two recent literature reviews: Peet et al. in the Economics of Education Review 2020, and Patrinos and Psacharopoulos in the Economics of Education (also 2020). I am (furthermore) additionally skeptical of this analysis because the authors don't have actual income data, but are imputing income based on assets as reported in the DHS.

My suggestion is that this entire analysis be dropped from the paper and that the authors focus only on the health benefits of secondary education. Here the authors seem to be on stronger ground in reporting only a correlation for several reasons: they are using a relatively novel data source with long-term longitudinal data; and the literature is smaller. However, the authors again need to do a lot more to situate their findings in this literature, however large it is. There are a number of papers cited (citations 3 to 10, broadly), but in the discussion section the authors should more clearly elaborate on what they find relative to the existing papers and what contribution these findings make. Are the existing estimates from comparable settings or using similar research designs, or are they different? Is the magnitude of the relationship between education and mortality they estimate similar to existing estimates, or different? Is the potential bias due to endogenous selection into education similar in this analysis, or worse? (In general - in a context such as rural Burkina Faso where almost no one attends secondary school, we would expect this bias to be particularly acute: only highly motivated and intelligent students from families with adequate resources and/or a high interest in education would attend secondary school.)

Having dropped the analysis of income returns to education, the authors would then have more space for a longer discussion section that might fruitfully grapple with some of these questions around situating the mortality analysis in the literature and unpacking how to interpret this relationship.

6. PLOS authors have the option to publish the peer review history of their article (what does this mean?). If published, this will include your full peer review and any attached files.

Reviewer #1: No

Reviewer #2: No

---

## [Author Response · Author response to Decision Letter 0]

6 Mar 2022

Response to Reviewers

Review Comments to the Author 

Reviewer #1: 

1) I think this is a very interesting paper on a very important topic. It is evident that there's a lot of hard work behind this manuscript however, in some places, it seems that the authors assumed that the crucial aspects of their analysis are common knowledge. For instance, Cox methodology is widely used but this paper should at least provide a brief description about it. Similarly, datasets and their sampling methodologies are not explained in detail. I think it is very important to include this information.

We thank the reviewer for his/her valuation of the work provided for this research. More detailed explanations on methodology and datasets were included, for example:

“To determine the relationship between education and survival status, we employed a multivariable Cox proportional hazards model. Our outcome was all-cause mortality. The Cox proportional hazards model is a survival model that allows the hazard of a certain event (here: death) to change over time under the condition that the hazard ratios in between the different groups stay the same (proportional-hazards assumption) (34). In our application, the Cox proportional hazards model describes the hazard of “all-cause mortality” (outcome) while assuming that the hazards in between the different schooling groups remain the same.”

2) From estimations it appears as if the variables reported in the tables are the only ones included in the regressions i.e there are no controls. This is potentially problematic and all confounding variables should be included. There are several demographic variables in DHS that could be used.

In the survival regressions, we controlled for sex and year of birth in the basic model. The Nouna HDSS dataset does not include too many other relevant controls for the survival analysis. 

With regard to the earnings regressions, all models included controls for age squared. We tried to include the most important pre-determined variables by stratifying for sex and type of location (urban/rural). Those stratified models additionally included controls for DHS survey round. We considered other pre-determined variables such as place of birth or birth order, but these are only available for younger household members in the DHS. We also considered other factors such as marital status or fertility but decided to exclude them from the model because these factors are likely directly determined by education and thus partially explain the effects of interest.

“We abstained from including more controls in the [Mincer]equation as this might exacerbate bias already existent in the data (38, 41.)”

3) Mincer’s equation also seems to be incomplete. Age squared in missing and authors should also attempt education squared in continuous form. See schooling locus in George Borjas book of labor economics.

We apologize for the lack of clarity. Age squared is one of the variables we had included in the Mincer regression, as recommended by the Reviewer, but was not shown in the main tables for visualization purposes. We have now further clarified this point throughout the paper:

“We adjusted for age, as a proxy for labor market experience, age squared as part of the Mincer equation and DHS survey year.”

“Table 5. Income analysis: Results from OLS earnings regression models for the Boucle du Mouhoun region, Burkina Faso.

[Table] 

Notes: […] In all models we controlled for age squared; in the stratified models we additionally controlled for survey round. OLS: ordinary least squares.”

 Additionally, we have now included regression results when adding education squared in continuous form in the supplementary information (shown in Table S3).

4) Citations seem a little unusual to me. Authors mention study number for direct citations and mention the sane number in parentheses as well. Please check the convention.

We thank the Reviewer for bringing this to our attention. We have now carefully reviewed the reference list and have corrected all corresponding citations.

5) It would've been awesome if authors explained the results in context of Burkina Faso. It would've been easier to grasp the meaning of 1000 USD in context of the country being studied. Storytelling is the weakest aspect of this paper. Too much emphasis is on estimations and even the review section seems to be rushed. However, with little polishing, it can be accepted.

We have now provided additional context for the results – e.g., see:

“We also show that the financial returns to secondary school are high, with an estimated additional USD 8,000 of lifetime income which equals almost 1,740,000 CFA representing around 3 to 4 annual salaries based on GDP per capita (44, 45). Combining health and economic benefits suggests a total return between approx. USD 10,000 and USD 33,000 (> 7,000,000 CFA) in this region.”

We decided that it would be difficult, however, to add more daily life comparisons - such as the rent of an apartment - as there is high price variability and few reliable sources to cite such comparisons.

Reviewer #2: 

1) This paper provides evidence about the health and economic benefits of secondary education in rural Burkina Faso, using two sources of evidence: long-run health surveillance data, and the DHS. They find evidence that attending secondary school is associated with higher income as well as higher life expectancy. My overriding comment about this paper is that it provides only correlational evidence about the relationship of interest, when we know that correlational evidence is probably severely limited, and causal evidence is widely available. The authors' acknowledgment of this fact (absence of causal evidence) is limited to a single sentence in the limitations section. Particularly for the analysis of the returns to education, there is an absolutely enormous economics literature analyzing causal effects of education on earnings in the developing world already published, and so the bar for an additional contribution is high. The authors fail to cite the majority of this literature, I won't aim to summarize it myself but will refer the authors to two recent literature reviews: Peet et al. in the Economics of Education Review 2020, and Patrinos and Psacharopoulos in the Economics of Education (also 2020). 

We thank the reviewer for this helpful comment on the pre-existing literature. We included the contribution in the Economics of Education Review 2020 from Patrinos and Psacharopoulos and also added some additional references (Peet et al 2015, Patrinos and Psacharopolous 2018).

“The results presented here build on recent research showing that average rates of return are high in general and especially in sub-Saharan Africa (e.g., 12,5% in Montenegro et al, 2014), and that adolescents enrolled in school have a wide range of improved health outcomes compared to their peers in this setting (4, 48, 49).”

2) I am (furthermore) additionally skeptical of this analysis because the authors don't have actual income data but are imputing income based on assets as reported in the DHS.

We acknowledge the fact that only imputed income is used in our research. Income data is not available in the DHS, unfortunately, and we have now further clarified this in the paper:

“Data were extracted on sex, age, education, household wealth assets, geographical region, sampling weights, date of interview, number of household members, type of place of residence (rural/urban), and marital status, using information from the DHS household recode, personal recode and individual recode files. The DHS did not collect information on income because a majority of the population in this low-resource setting engages in subsistence farming or depend on informal jobs with irregular incomes.”

3) My suggestion is that this entire analysis be dropped from the paper and that the authors focus only on the health benefits of secondary education. Here the authors seem to be on stronger ground in reporting only a correlation for several reasons: they are using a relatively novel data source with long-term longitudinal data; and the literature is smaller. Having dropped the analysis of income returns to education, the authors would then have more space for a longer discussion section that might fruitfully grapple with some of these questions around situating the mortality analysis in the literature and unpacking how to interpret this relationship. 

We agree with the Reviewer that the economic returns to education literature is large and that we only report correlational data. There are other papers that look at survival benefits as well, but to our knowledge there is no paper that has tried to combine economic and health returns in this context, which is in our view the novelty of this paper.

4) However, the authors again need to do a lot more to situate their findings in this literature, however large it is. There are a number of papers cited (citations 3 to 10, broadly), but in the discussion section the authors should more clearly elaborate on what they find relative to the existing papers and what contribution these findings make. Are the existing estimates from comparable settings or using similar research designs, or are they different? Is the magnitude of the relationship between education and mortality they estimate similar to existing estimates, or different? Is the potential bias due to endogenous selection into education similar in this analysis, or worse? (In general - in a context such as rural Burkina Faso where almost no one attends secondary school, we would expect this bias to be particularly acute: only highly motivated and intelligent students from families with adequate resources and/or a high interest in education would attend secondary school.)

We agree that the literature on returns to education is large – estimates from this setting are however rare. There are other papers that look at survival benefits as well, but to our knowledge there is no paper that has tried to combine them, which is in our view the novelty of this paper. We would be most grateful for any additional references if there are any.

We concur that endogenous selection into education is possible, which would affect health and economic outcomes in similar ways and therefore not falsify our comparisons between health and economic returns. Literature on returns to education, however, suggests that instrumental variables estimates are generally similar to - or slightly larger - than ordinary least squares based Mincerian regressions (1). This is why we consider selection biases not being large. We have now tried to include more literature to better situate our findings:

“The results presented here build on recent research showing that average rates of return are high in general and especially in sub-Saharan Africa (e.g. 12,5% in Montenegro et al, 2014), and that adolescents enrolled in school have a wide range of improved health outcomes compared to their peers in this setting (4, 48, 49). Perceived returns to education are high among other key stakeholders in the demographic surveillance area, such as parents, teachers, as well as health workers (50). There appears to be a general consensus in literature that women receive higher economic benefits than men (for example an average rate of return of 14,6% for women vs, 11,3% for men) (4, 8, 50, 51) in line with our findings. Peet et al, (2015) found lower benefits for rural areas contrary to our findings. Concerning returns to schooling according to schooling level, Barouni and Broecke (2014) found higher returns for higher (secondary and tertiary) schooling than for primary schooling similar to the current paper, while Psacharopoulos (1985) found highest returns for primary education in the past. This might show a tendency towards higher returns for higher schooling levels. Concerning the methodology, most studies on returns to schooling use the discounting method or Mincer regression similar to us (49). 

Literature on returns to education is large – estimates from this setting are however rare (54, 55). Most of these studies, however, focus on either survival (3) or economic benefits (56, 57). To our knowledge, only one study to date examines both monetary and non-pecuniary returns to schooling, but in a high-resource setting (58).”

Literature Cited

1. Patrinos HA, Psacharopoulos G. Returns to education in developing countries. In: The Economics of Education: Elsevier; 2020. p. 53–64.

---

## [Editor Report · Decision Letter 1]

14 Apr 2022

PONE-D-21-01735R1Health and economic benefits of secondary education in the context of poverty: Evidence from Burkina FasoPLOS ONE

Dear Dr. Werner,

Thank you for submitting your manuscript to PLOS ONE. After careful consideration, we feel that it has merit but does not fully meet PLOS ONE’s publication criteria as it currently stands. Therefore, we invite you to submit a revised version of the manuscript that addresses the points raised during the review process.

I have now been asked to serve as a guest editor for this submission; in the interests of transparency, I should note I was previously serving as a reviewer (reviewer #2).  I reviewed the updated materials you provided and also invited reviewer #1 to review the manuscript again; s/he declined to do so. My judgment is that you have responded thoroughly to this reviewer's comments.

Following up on my own comments, I would like to request some minor additional revisions.

-I am uncomfortable with the use of imputed income in an analysis of returns to education, though of course you are right that the DHS does not report income. I suggest that you conduct this analysis primarily using an asset index drawing on the asset information that is actually reported in the DHS (i.e., estimate the correlation between education and assets.) Since this is not in any case a causal analysis that would fall into the core returns to education literature, it does not seem necessary to use an artificial income measure. If you wish to also include the analysis using imputed income as an extension in the paper, of course you can, but I suggest this be a secondary analysis.

-In the discussion section, you focus primarily on comparing your estimate of the returns to education to other estimates. What about other estimates of mortality benefits of education - are there any such estimates? What is the magnitude of those estimates compared to yours? It may be that the literature here is minimal; if so, you can just note this.

-Following up on your response to the previous referee report: it is not necessarily the case that bias in the estimated returns to education due to selection into education is the same across the two dimensions examined (income and mortality). For example, if very intelligent individuals (who are not necessarily healthy or rich) select into education, the selection bias may be larger for income vis-à-vis mortality. If individuals from wealthy families select into education, then the bias may be inverted (larger for mortality, since wealthier families live longer, ceteris paribus). It is up to you whether you wish to engage with this point in the discussion section, but it may be useful context for your interpretation.

We look forward to receiving your revised manuscript.

Kind regards,

Jessica Leight, PhD

Academic Editor

PLOS ONE
---

## [Author Response · Author response to Decision Letter 1]

24 May 2022

1) I have now been asked to serve as a guest editor for this submission; in the interests of transparency, I should note I was previously serving as a reviewer (reviewer #2). I reviewed the updated materials you provided and also invited reviewer #1 to review the manuscript again; s/he declined to do so. My judgment is that you have responded thoroughly to this reviewer's comments. Following up on my own comments, I would like to request some minor additional revisions. 

We thank the reviewer for his/her judgment of this paper and the work invested in a second revision.

2) I am uncomfortable with the use of imputed income in an analysis of returns to education, though of course you are right that the DHS does not report income. I suggest that you conduct this analysis primarily using an asset index drawing on the asset information that is actually reported in the DHS (i.e., estimate the correlation between education and assets.) Since this is not in any case a causal analysis that would fall into the core returns to education literature, it does not seem necessary to use an artificial income measure. If you wish to also include the analysis using imputed income as an extension in the paper, of course you can, but I suggest this be a secondary analysis.

We replaced the income estimates with the original asset score in form of asset quintiles and used ln(asset quintile) as dependent variable to calculate increase in asset holdings and described health and economic benefits separately. 

“We use asset ownership as a proxy for permanent income and estimate average increases in asset holdings associated with additional education. Asset quintiles were used as outcome in standard Mincer regression models to estimate returns to education (38–40).”

We moved the predicted income table to the supporting information (Supplementary Analysis S1) as suggested.

“For methodology and estimation of predicted incomes and relative health and financial returns see Supplementary Analysis S1 in the supporting information.”

3) In the discussion section, you focus primarily on comparing your estimate of the returns to education to other estimates. What about other estimates of mortality benefits of education - are there any such estimates? What is the magnitude of those estimates compared to yours? It may be that the literature here is minimal; if so, you can just note this.

There is indeed not nearly as much literature on the survival returns to education - we have now added the two main references to the revised Discussion section, where we write: 

“Concerning the magnitude of health benefits, Gathmann, Jürges et al. (2015) estimate a 2.0% reduction in 20-year-mortality rate for only men (9) and Lleras-Muney (2005) a 10-year mortality rate reduction by 1.3 to 3.6% per additional year of education (47). Those results are similar to our estimations finding a 3.6% reduction in the mortality hazard.”

4) Following up on your response to the previous referee report: it is not necessarily the case that bias in the estimated returns to education due to selection into education is the same across the two dimensions examined (income and mortality). For example, if very intelligent individuals (who are not necessarily healthy or rich) select into education, the selection bias may be larger for income vis-à-vis mortality. If individuals from wealthy families select into education, then the bias may be inverted (larger for mortality, since wealthier families live longer, ceteris paribus). It is up to you whether you wish to engage with this point in the discussion section, but it may be useful context for your interpretation.

We thank the reviewer for bringing this to our attention; we have now added the following point to the revised Discussion section:

“Fifth, endogenous selection bias cannot be excluded: Wealthy individuals for example might select into education and bias health and income benefits positively, or very intelligent individuals who might bias stronger income than mortality results.”

---

## [Editor Report · Decision Letter 2]

8 Jun 2022

Health and economic benefits of secondary education in the context of poverty: Evidence from Burkina Faso

PONE-D-21-01735R2

Dear Dr. Werner,

We’re pleased to inform you that your manuscript has been judged scientifically suitable for publication and will be formally accepted for publication once it meets all outstanding technical requirements.

Kind regards,

Jessica Leight, PhD

Guest Editor

PLOS ONE

Additional Editor Comments (optional):

Thanks so much for submitting the revised manuscript, Health and economic benefits of secondary education in the context of poverty: Evidence from Burkina Faso. I'm happy to accept your manuscript for publication in PloS One and believe it has the potential to make a significant contribution to the literature.

---

## [Editor Report · Acceptance letter]

14 Jun 2022

PONE-D-21-01735R2 

Health and economic benefits of secondary education in the context of poverty: Evidence from Burkina Faso 

Dear Dr. Werner:

I'm pleased to inform you that your manuscript has been deemed suitable for publication in PLOS ONE. Congratulations! Your manuscript is now with our production department. 

Kind regards, 

on behalf of

Dr. Jessica Leight 

Guest Editor

PLOS ONE